# Clinical Value and Molecular Function of Circulating MicroRNAs in Endometrial Cancer Regulation: A Systematic Review

**DOI:** 10.3390/cells11111836

**Published:** 2022-06-03

**Authors:** Joy Bloomfield, Michèle Sabbah, Mathieu Castela, Céline Mehats, Catherine Uzan, Geoffroy Canlorbe

**Affiliations:** 1Cancer Biology and Therapeutics, Centre de Recherche Saint-Antoine (CRSA), Sorbonne University, INSERM UMR_S_938, 75020 Paris, France; joybloomf@gmail.com (J.B.); michele.sabbah@inserm.fr (M.S.); catherine.uzan@aphp.fr (C.U.); 2Assistance Publique des Hôpitaux de Paris (AP-HP), Department of Gynecological and Breast Surgery and Oncology, Pitié-Salpêtrière University Hospital, 75013 Paris, France; 3Centre National de la Recherche Scientifique (CNRS), 75012 Paris, France; 4Scarcell Therapeutics, 101 Rue de Sèvres, 75006 Paris, France; castela.mathieu@yahoo.fr; 5U1016, CNRS, UMR8104, Institut Cochin, INSERM, Université de Paris, 75014 Paris, France; celine.mehats@inserm.fr; 6Institut Universitaire de Cancérologie (IUC), 75020 Paris, France

**Keywords:** microRNA, miR, miRNA, endometrial cancer, biomarkers, circulating, plasma, miR-27a, miR-29b, miR-150-5p

## Abstract

This systematic review of literature highlights the different microRNAs circulating in the serum or plasma of endometrial cancer patients and their association with clinical and prognostic characteristics in endometrial cancer. This study also investigates the molecular functions of these circulating microRNAs. According to this systematic review, a total of 33 individual circulating miRs (-9, -15b, -20b-5p, -21, -27a, -29b, -30a-5p, -92a, -99a, -100, -135b, -141, -142-3p, -143-3p, -146a-5p, -150-5p, -151a-5p, -186, -195-5p, -199b, -200a, -203, -204, -205, -222, -223, -301b, -423-3p, -449, -484, -887-5p, -1228, and -1290) and 6 different panels of miRs (“miR-222/miR-223/miR-186/miR-204”, “miR-142-3p/miR-146a-5p/miR-151a-5p”, “miR-143-3p/miR-195-5p/miR-20b-5p/miR-204-5p/miR-423-3p/miR-484”, “mir-9/miR-1229”, “miR-9/miR-92a”, and “miR-99a/miR-199b”) had a significant expression variation in EC patients compared to healthy patients. Also, seven individual circulating miRs (-9, -21, -27a, -29b, -99a, -142-3p, and -449a) had a significant expression variation according to EC prognostic factors such as the histological type and grade, tumor size, FIGO stage, lymph node involvement, and survival rates. One panel of circulating miRs (“-200b/-200c/-203/-449a”) had a significant expression variation according to EC myometrial invasion. Further studies are needed to better understand their function and circulation.

## 1. Introduction

Endometrial cancer (EC) is the most frequent gynecologic cancer in developed countries with an increasing incidence due to longer life expectancy and increasing obesity [1]. Its incidence in developed countries is 11.1 per 100,000 women compared to 3.3 per 100,000 women in developing countries [2].

Although the mechanisms underlying endometrial carcinogenesis are not fully understood, current evidence suggests that alterations in the epigenome result in both the expression of oncogenes and the downregulation of tumor suppressors, therefore promoting tumor initiation and progression in EC [3].

MicroRNAs (miR) are a family of small non-coding RNAs measuring 21–25 nucleotides in length that are involved in epigenetic mechanisms. Each miR has the potential to regulate a variety of genes (usually about 500), and each gene is usually targeted by several different miRs [4].

MiRs have been linked to carcinogenesis and can act as activators or suppressors of tumors. Therefore, they can potentially be used as diagnostic and prognostic factors. MiR expression in EC tissues has been extensively studied and has been proven as a potential biomarker [5]. This study concentrates on plasmatic miR expression in EC patients as potential biomarkers for diagnosis and prognosis. The molecular function of the miRs responsible for these clinical implications will also be explored in this study.

The main objective of this study is to present the clinical contributions of circulating miRs in therapeutic management, including the correlation to histological type, FIGO stage, lymph node status, and overall disease-free survival. At the same time, the objective will be to explain the molecular functions underlying these clinical contributions.

## 2. Materials and Methods

Our systemic review for circulating miR expression in EC was carried out by using the following databases and following the Preferred Reporting Items for Systematic Reviews and Meta-Analyses (PRISMA) guidelines (Figure 1): PubMed (the Internet portal of the National Library of Medicine), MEDLINE, and the Cochrane database.

The PRISMA 2020 checklist is reported in Appendix A.

We used the following terms: “microRNA”, “miR”, “miRNA”, “endometrial cancer”, “plasma”, “serum”, “circulating”, and “biomarkers”. No filters or limits were used.

We included only articles concerning plasmatic or serum miRs in EC that were published in English between 1 May 2009 and 31 November 2021.

Articles in which miRs samples were exclusively obtained from primitive EC tissues (hysterectomy) or other liquid biopsies (urine, saliva, intraoperative ascites) were excluded. Articles that did not mention EC and were not in English were also excluded.

The studies were grouped per type (prospective study, systematic review, and meta-analysis) and by objective for synthesis.

Each article was initially screened by title by the same reviewer (J.B.). The articles were sought for retrieval by abstract and for eligibility by the lecture of the article. These were read by two independent reviewers (J.B. and G.C.). If there existed a disagreement on an article, the selection was made collectively.

The data was collected by the initial reviewer (J.B.) and was checked by the second reviewer (G.C.). Information concerning the miRs in each article was then categorized according to their expression levels in serum/plasma, diagnostic performance, and clinical value according to known prognostic factors [6]: histological type (endometrioid and non-endometrioid), FIGO stages (classified I through IV), histopathological grades (1, 2 and 3), lymphovascular space invasion (LVSI) (positive or not), lymph node metastasis (LNM) (positive if one lymph node was positive, including both macrometastases and micrometastases <2 mm), and overall survival rate (calculated in months). The method of analysis of the miRs in each article was also collected: extraction kit, conservation (temperature), microarray kit, and PCR kit. Last of all, we evaluated the molecular function of the circulating miRs in their extracellular and intracellular compartments.

If information was missing, the article was excluded and no assumptions were made. A bias assessment was not applicable in this systematic review. Each study was detailed independently (type of study, number of patients included, and types of patients included) in order to provide the reader with the information necessary for the evaluation of their quality.

Our systematic review was registered in the International Prospective Register of Systematic Reviews (PROSPERO): CRD42022303963.

## 3. Results

According to the PRISMA guidelines for the study selection process, we found a total of 28 articles concerning plasmatic miR in EC: 14 were original articles, 1 was an original article with a meta-analysis, 2 were meta-analyses, 2 were systematic reviews, and 9 were reviews (Table 1).

### 3.1. Clinical Value

#### 3.1.1. Circulating miRs in EC Patients Compared to Healthy Patients without EC

##### Diagnostic Performance

A total of 15 articles evaluated the diagnostic performance of circulating miRs (serum, plasmatic, and venous blood) to detect EC (7–21) (Table 2).

A total of 33 individual circulating miRs had a significant variation of expression in EC patients compared to healthy patients: miR-9, miR-15b, miR-20b-5p, miR-21, miR-27a, miR-29b, miR-30a-5p, miR-92a, miR-99a, miR-100, miR-135b, miR-141, miR-142-3p, miR-143-3p, miR-146a-5p, miR-150-5p, miR-151a-5p, miR-186, miR-195-5p, miR-199b, miR-200a, miR-203, miR-204, miR-205, miR-222, miR-223, miR-301b, miR-423-3p, miR-449, miR-484, miR-887-5p, miR-1228, and miR-1290.

Among these 33 miRs, 27 miRs were overexpressed in the plasma/serum of EC patients compared to healthy patients: miR-15b, miR-20b-5p, miR-27a, miR-92a, miR-99a, miR-100, miR-135b, miR-141, miR-142-3p, miR-143-3p, miR-146a-5p, miR-150-5p, miR-151a-5p, miR-186, miR-195-5p, miR-199b, miR-200a, miR-203, miR-205, miR-222, miR-223, miR-423-3p, miR-449a, miR-484, miR-887-5p, miR-1228, and miR-1290.Among these miRs, 3 miRs had the best diagnostic performance: miR-205 [11] had an AUC of 1.0 (95% IC: 1.000–1.000); miR-27a [20] was upregulated in the plasma of EC patients compared to patients without EC and had an AUC of 1.000 (*p* < 0.001) with a sensitivity and specificity of 100% and 100%, respectively, and a positive predictive value and a negative predictive value of 100% and 100%; and miR-150-5p [20] was upregulated in the plasma of EC patients compared to patients without EC and had an AUC of 0.982 (*p* < 0.001) with a sensitivity and specificity of 88.89% and 100%, respectively, and a positive predictive value and a negative predictive value of 100% and 78.9%. As noted in Table 2, these studies were based on a small number of patients (12 EC patients and 12 healthy patients for miR-205 and 36 EC patients and 36 healthy patients for miR-27a and miR-150-5p).Among these 33 miRs, 4 were under-expressed in the plasma/serum of patients with EC compared to healthy patients: miR-9, miR-29b, miR-30a-3p, and miR-301b. Among them, 1 miR had the best diagnostic performance: miR-29b [15] had an AUC of 0.976 (95% IC: 0.951–1.000) with a cutoff value of 0.940 and with a sensitivity of 96.1% and specificity of 97.9%. As noted in Table 2, this study was based on comparing 356 EC patients to 155 healthy patients. This miR was significantly lower in EC patients and had the same ability to discriminate EC patients from healthy patients whether the EC patients were metastatic or not [15]. MiR-29b also had the particularity to be able to discriminate EC patients from healthy patients and from patients with benign endometrial lesions (polyps, myomas): miR-29b expression remained significantly lower (*p* < 0.05) in patients with EC (0.893 ± 0.432) compared to healthy patients (1.070 ± 0.130) and patients with benign uterine lesions (1.036 ± 0.112) [15].Two miRs had various expression levels in the plasma/serum according to different studies (miR-21 and miR-204). Among them, 1 miR had the best diagnostic performance: miR-204 [13] had an AUC of 1.000 (95% IC: 1.000–1.000) with a sensitivity and specificity of 100% each. In this study, miR-204 was downregulated. As noted in Table 2, this study was based on a small number of patients (46 EC patients and 28 healthy patients). Two other studies [8,17] found that miR-204 was upregulated in the serum of patients with EC with a lower diagnostic value with an AUC of 0.740 (95% IC: 0.594–0.885) and 0.668 (95% IC: 0.592–0.743). In one study [21], miR-21 was able to discriminate EC patients from healthy patients and from patients with benign endometrial lesions (polyps, myomas) with a diagnostic performance for EC with an AUC of 0.831 (95% IC: 0.746–0.916) with a sensitivity and specificity of 70% and 92%, respectively, for a cutoff value of 2.937 compared to healthy patients. Healthy patients had an AUC of 0.710 (95% IC: 0.608–0.813) with a sensitivity and specificity of 64% and 76%, respectively, for a cutoff value of 3.457 compared to patients with benign uterine lesions.

A total of six signatures of miRs were also evaluated in five studies for their ability to discriminate EC with healthy patients [7,8,9,17,19]: “miR-222/miR-223/miR-186/miR-204”, “miR-142-3p/miR-146a-5p/miR-151a-5p”, “miR-143-3p/miR-195-5p/miR-20b-5p/miR-204-5p/miR-423-3p/miR-484”, “mir-9/miR-1229”, “miR-9/miR-92a”, and “miR-99a/miR-199b”.

Among them, the miR signature with the best diagnostic performance was “miR-222/miR-223/miR-186/miR-204” [8] with an AUC of 0.927 (95% IC: 0.845–1.000) and a sensitivity of 91.7% and a specificity of 87.5%. As noted in Table 2, this study was based on a small number of patients (26 EC patients and 22 healthy patients).The diagnostic performance of the 6-miR signature “miR-20b-5p/miR-143-3p/miR-195-5p/miR-204-5p/miR-423-3p/miR-484” [17] and the 3-miR signature “miR-142-3p/miR-146a-5p/miR-151a-5p” [19] remained significant in the diagnostic of EC compared to healthy patients when sub-categorizing the EC patients within their FIGO stage (I and II–IV) or within their histological grade (G1, G2, and G3).

One study evaluated the association of miR expression with other markers. A combination of miR-27a and CA 125 was evaluated in the study of Wang et al. [10] and had an AUC of 0.894 (95% CI 0.807–0.980) with a sensitivity of 77.4% and a specificity of 97%.

##### Prognosis

The miRs associated with prognostic factors in EC patients compared to healthy patients are detailed in the Table 3.

##### Grade

A total of 14 miRs had a significant variation of expression when comparing different histological grades in EC patients compared to healthy patients. When comparing EC with histological G1 to healthy patients, *miR-9, miR-92a, miR-141, miR-200a, miR-203, miR-449, miR-1228, miR-1290, miR-143-3p, miR-195-5p, miR-20b-5p, miR-204-5p, miR-423-3p*, and *miR-484* expression were significantly different [9,17]. When comparing EC with histological G2 to healthy patients, *miR-143-3p, miR-195-5p, miR-20b-5p, miR-204-5p,* and *miR-484* expression were significantly different [17]. When comparing EC with histological G3 to healthy patients, *miR-20b-5p, miR-143-3p, miR-195-5p, miR-423-3p,* and *miR-484* expression were significantly different [17]. When comparing EC with histological G2–G3 to healthy patients, *miR-9, miR-92a, miR-141, miR-200a, miR-449a, miR-1228,* and *miR-1290* expression were significantly different [9].A total of four miRs (-203, -204-5p, 301b, 423-3p) did not have a significant variation of expression when comparing different histological grades in EC patients compared to healthy patients [9,17].

##### FIGO

A total of 19 miRs had a significant variation of expression within different FIGO stages of EC patients compared to healthy patients. When comparing miR expression in patients with FIGO stage I EC to healthy patients, miR-186, miR-222, miR-223, miR 204, miR-143-3p, miR-195-5p, miR-20b-5p, miR-423-3p, and miR-484 expression levels were significant [13,17]. Furthermore, the expression levels and diagnostic performance of miR-186, miR-204, miR-222, and miR-223 remained significant with an AUC of 0.73 (*p* = 0.002), 1.00 (*p* < 0.0001), 0.71 (*p* = 0.006), and 0.85 (*p* < 0.0001), respectively [13]. When comparing miR expression in patients with FIGO stage II–IV EC to healthy patients, miR-143-3p, miR-195-5p, miR-20b-5p, miR-423-3p, and miR-484 expression levels were significant [17]. When comparing miR expression in patients with FIGO stage IA EC to healthy patients, miR-9, miR-92a, miR-99a, miR-141, miR-199b, miR-203, miR-449a, miR-1228, and miR-1290 expression levels were significant [7,9]. When comparing miR expression in patients with FIGO stages > IA EC to healthy patients, miR-9, miR-92a, miR-99a, miR-100, miR-141, miR-199b, miR-200a, miR-203, miR-449a, miR-1228, and miR-1290 expression levels were significant [7,9].A total of four miRs (-100, -200a, 204-5p, 301b) did not have a significant variation of expression within different FIGO stages of EC patients compared to healthy patients [7,9,17].

#### 3.1.2. Prognosis in EC Patients

A total of 25 miRs were analyzed for their association to prognostic factors between EC patients. These miRs are detailed in Table 4.

##### Histological Type

A total of two studies evaluated miR expression in venous blood and its association to the histological type of EC [15,20]. *MiR-27a* was significantly overexpressed in EC patients with endometrioid carcinoma compared to patients with special types (28 patients with endometrioid carcinoma and 8 patients with special types). There was not a significant difference in *mir-29b* (150 patients with endometrioid carcinoma and 25 patients with special types) and *miR-150-5p* (28 patients with endometrioid carcinoma and 8 patients with special types) expression regarding the histological type of the tumor.

##### Histological Grade

A total of nine studies evaluated peripheral miR expression (serum, plasma) and its association to the histological grading of EC [7,9,11,13,14,15,17,19,20].

A total of three miRs had a variation of expression within different histological grades of patients with EC. *MiR-142-3p* and *miR-21* levels were higher in patients with histological grade 1 (G1) EC [11,19]. *MiR-9* was overexpressed in histological grade 2 (G2) and grade 3 (G3) compared to G1 EC patients [9].

MiR expression levels within different histological groups were not significantly different for *miR-29, miR-30a-3p, miR-92, miR-135b, miR-141, miR-200a, miR-203, miR-205, miR-301b, miR-449, miR-1228,* and *miR-1290* (G2–G3 compared to G1) [9,11,15]; *miR-186, miR-222,* and *miR-223* (G3 vs. G1–G2) [13]; and *miR-27a, miR-99a, miR-100, miR-150-5p,* and *miR-199b* (G1, G2 and G3) [7,20].

##### Primitive Tumor Size

One study evaluated peripheral miR expression and its association to tumor size in patients with EC [15]. Patients with tumor sizes >6 cm had a significantly lower level of *miR-29b* expression than patients having tumor sizes ≤6 cm; they were 0.664 ± 0.443 and 1.090 ± 0.702 (*p* = 0.014) for EC patients without metastasis, respectively, and 0.632 ± 0.365 and 0.894 ± 0.480 (*p* < 0.001) for EC patients without metastasis, respectively.

##### Myometrial Invasion

A total of four studies evaluated the circulating miR expression (venous blood, plasma, and serum) and its association with myometrial invasion in EC patients [7,9,15,20].

Endometrial tumors invading ≥50% can be discriminated from endometrial tumors invading less than half of the myometrium when combining four miRs (*miR-200b/miR-200c/miR-203/miR-449a)* with an AUC of 0.851 (95% CI: 0.687–0.949) [9].

MiR expression levels concerning different myometrial invasion were not significantly different for miR-9, miR-27a, miR-29b, miR-92a, miR-99a, miR-100, miR-141, miR-150-5p, miR-199b, miR-200a, miR-203, miR-301b, miR-449a, miR-1228, and miR-1290 [7,9,15,20].

##### FIGO Stage

A total of eight studies evaluated peripheral miR expression (venous blood, plasma, and serum) and its association with different FIGO stages of EC [7,9,11,14,15,17,19,20].

A total of four miRs had a variation of expression within different FIGO stages of patients with EC. *MiR-29b* expression in EC patients’ blood samples was correlated ith FIGO staging and was significantly lower in FIGO III and IV stages than in FIGO I and II stages (*p* < 0.05) [15]. The plasma concentration of *miR-21* was significantly higher in EC patients with a FIGO stage of IA (*p* = 0.017) compared to EC patients with FIGO stages ≥IB [11]. When comparing miR expression in EC patients with FIGO stage IA to FIGO stages >IA, *miR-99a* and *miR-449a* expression levels were significantly different (7,9).

MiR expression levels within different FIGO stages were not significantly different for miR-9, miR-27a, miR-30a, miR-92a, miR-100, miR-135p, miR-141, miR-150-5p, miR-199a, miR-200a, miR-203, miR-205, miR-301b, miR-1228, miR-1290 (FIGO stage IA, FIGO stages >IA), miR-142-3p, miR-146a-5p, and miR-151a-5p (FIGO stage I, FIGO stages II–IV) [7,9,19].

##### Lymph Node Metastasis

A total of two studies evaluated peripheral miR expression (venous blood) and its association to LNM in EC patients [15,20]. EC patients with LNM (98 patients) had significantly lower *miR-29b* expression levels than patients without LNM (266 patients); they were 0.654 ± 0.453 compared to 0.988 ± 0.669 (*p* = 0.011) for EC patients without LNM, respectively, and 0.481 ± 0.370 compared to 0.855 ± 0.0.435 (*p* < 0.001) for EC patients with LNM, respectively [15]. There was an absence of statistical difference in the expression of *miR-27a* and *miR-150-5p* in EC patients with LNM (6 patients) compared to EC patients without LNM (27 patients) [20].

##### Lymphovascular Space Invasion

One study evaluated plasmatic miR expression and its correlation to LVSI in EC patients [20]. *MiR-27a* and *miR-150-5p* expression was not statistically different in EC patients with LVSI (23 patients) compared to patients without LVSI (11 patients).

##### Distant Metastasis

No studies evaluated the plasmatic miR expression and its correlation with distant metastasis in EC patients

##### Average Survival Rate

One study evaluated peripheral miR expression (venous blood) and its association with survival in EC patients [15]. In EC patients without metastasis, there was a difference in survival rates according to the expression levels of *miR-29b*: an average survival period of 40.9 ± 1.3 months was observed in patients with low *miR-29b* expression compared to 54.6 ± 1.5 months in patients with high miR-29b expression. This difference of survival periods according to *miR-29b* expression levels was also observed in EC patients with metastasis: an average survival period of 35.5 ± 1.4 months was observed in patients with low *miR-29b* expression compared to 45.2 ± 1.9 months in patients with high *miR-29b* expression. This showed that patients with lower *miR-29b* levels had a shorter survival period (*p* < 0.05) and remained significant after multivariate analysis (*p* = 0.003 for patients without metastasis and *p* = 0.028 for patients with metastasis) [15].

### 3.2. Molecular Function of Circulating miRs in EC

An explanatory scheme of the different mechanisms of miR molecular functions has been presented in Figure 2.

MiRs have been detected in plasma and other various fluids (saliva, urine). The function of circulating miRs remains poorly understood. There are essentially two theories: (1) the passive release of miRs in body fluids resulting after apoptosis and/or another cellular activity and (2) the active release of miRs that are discharged into the circulation by cells as messengers [22].

Even though RNases are present, miRs are relatively stable in blood. They are also stable in extreme conditions (temperature and pH variations) [22]. The stability of these extracellular miRs may be explained by different sorts of carriers:**Bound to Argonaute (AGO) proteins**: AGO-protein-bounded miRs are the largest form of extracellular circulation and represent up to 90–95% of the circulating miRs that are found in plasma. The AGO protein binds with the miR in the intracellular compartment in order to create the RISC-complex, which regulates ARN messenger expression by cleavage or translational interference. It is this same AGO-protein–miR complex that is found in the extracellular compartment, either alone or within a micro-vesicle or an HDL-particle [22].**Encapsulated in micro-vesicles**, such as exosomes. It remains uncertain whether miRs are always bound to an AGO protein inside these micro-vesicles or not. The circulation of miRs in exosomes can result from either a passive or an active secretion from the tumor cell. Micro-vesicular miRs may represent the smallest fraction of circulating miRs [22].

In the study of Fan et al. [20], three miRs (*miR-142-3p, miR-146a-5p,* and *miR-151a-5p*) were overexpressed in the plasma of EC patients compared to healthy patients, whereas only *miR-151a-5p* was found in exosomes. This suggests that various miRs have different methods of circulation in the plasma. The same observation was made in a study of the same group: six miRs (*miR-143-3p, miR-195-5p, miR-20b-5p, miR-204-5p, miR-423-3p)* and *miR-484*) were overexpressed in the plasma of EC patients compared to healthy patients, but only *miR-20b-5p* was overexpressed in exosomes [18].

**Bound in High-density lipoproteins (HDL particles):** miR stability could also be explained by the fact that they circulate in HDL-particles. It is also unknown if miRs circulate when bound to an AGO protein within the HDL-particles or not, and if its secretion in the extracellular compartment is active or passive [22].**Apoptotic bodies**: based on the theory that miR secretion could be a passive mechanism resulting in tumor cell waste, miRs could also circulate in apoptotic bodies [22].

No data is currently available on the specific mechanism of miR release in the circulation in EC.

Three studies analyzed the molecular functions/regulation of certain miRs in relation with EC pathogenesis [7,18,20]. In a study of Torres et al. [7], miRs in EC tissues were studied in parallel with plasmatic miRs. The three miRs (*miR-99a, miR-100,* and *miR-199b*) that were overrepresented in the plasma were underrepresented in EC tissues compared to healthy patients. The authors linked the miRs levels to the expression of mTOR, a key regulator of cellular differentiation, proliferation, and reaction to stress that is overexpressed in EC tissues as compared to healthy samples.

In the two studies of Fan and al. [18,20], the DIANA-mirPath v3.0 database was used to identify the potential targets of miRs. At least 31 different pathways were found to be potentially affected from the deregulation of *miR142-3p, miR-146a-5p,* and *miR-151a-5p*, including Fc-epsilon receptor, TRI-dependent toll-like receptor, and MyD88-independent toll-like receptor signaling pathways. *MiR-143-3p, miR-195-5p, miR-20b-5p, miR-204-5p, miR-423-3p,* and *miR-484* were associated with numerous tumor-related pathways that are implicated in cell protein modification; nucleic acid binding transcription factors; and cytoskeleton proteins, fatty acid metabolism, the cell cycle, and p53 pathways. The overexpression of p53 is associated with a bad prognosis as it stimulates EC progression by targeting the proteasome activator REGγ. It is also often associated with higher histopathologic grades and lymph node metastasis in EC.

## 4. Discussion

According to this systematic literature review, a total of 33 individual circulating miRs (-9, -15b, -20b-5p, -21, -27a, -29b, -30a-5p, -92a, -99a, -100, -135b, -141, -142-3p, -143-3p, -146a-5p, -150-5p, -151a-5p, -186, -195-5p, -199b, -200a, -203, -204, -205, -222, -223, -301b, -423-3p, -449, -484, -887-5p, -1228, and -1290) and 6 different panels of miRs (“miR-222/miR-223/miR-186/miR-204”, “miR-142-3p/miR-146a-5p/miR-151a-5p”, “miR-143-3p/miR-195-5p/miR-20b-5p/miR-204-5p/miR-423-3p/miR-484”, “mir-9/miR-1229”, “miR-9/miR-92a”, and “miR-99a/miR-199b”) had a significant expression variation in EC patients compared to healthy patients. Additionally, seven individual circulating miRs (-9, -21, -27a, -29b, -99a, -142-3p, and -449a) had a significant variation of expression according to EC prognostic factors, such as the histological type and grade, tumor size, FIGO stage, lymph node involvement, and survival rates. One panel of circulating miRs (“-200b/-200c/-203/-449a”) had a significant variation of expression according to EC myometrial invasion.

The use of miRs in EC diagnosis could simplify the current approaches that are based on an endometrial biopsy in consultation or in the operating room under general anesthesia. The use of miRs could help in the detection of early-stage EC for at-risk patients and therefore allow an efficient and localized treatment. In this study, five miRs (-27a, 29b, 150-5p, -204 and -205) and one miR signature (“-20b-5p/-143-3p/-195-5p/-204-5p/-423-3p/-484”) seem to be the most significantly associated to EC diagnosis. Among them, circulating miR-29 has been shown to be effective in the diagnosis of colorectal cancer (CRC) with an AUC of 0.800 (CI 95%: 0.760–0.830) and a sensitivity and specificity of 65% and 82%, respectively [35]. Exosomal miR-27a has also been evaluated and associated with the diagnosis colorectal cancer (CRC). In a study by Liu et al. [36], miR-27a was significantly overexpressed in patients with CRC and had a diagnostic performance with an AUC of 0.773 (0.742) in the training phase, 0.82 (0.787) in the validation phase, and 0.746 (0.697) in the external validation phase. In this same study [36], the molecular function of miR-27a was explored and showed its association with Wnt/β-catenin and TGFβ pathways. Another miR has been evaluated for its diagnostic performance in breast cancer (BC): miR-205 expression is significantly downregulated in the serum of patients with breast cancer and has an AUC of 0.84 (CI 95%: 0.77–0.91) with a sensitivity and specificity of 86.2% and 82.8%, respectively [37]. It has been suggested that miR-205 promotes tumorigenesis by controlling the epithelial to mesenchymal transition by regulating SIP1 and ZEB1 pathways [38].

Concerning the prognosis of EC, the current initial surgical management is based on pre-operative histology and imaging. Recently, a molecular classification has been added to this management [39]. However, it is known that there is a discrepancy of risk groups between the pre-operative and the final classification, with a risk of over-treatment in 10% of cases and under-treatment in 37% of cases [40]. MiRs can help in pre-operative management by being associated with certain prognostic factors such as FIGO stage, LNM, LVSI, and survival rates. In the study of Delangle et al. [41], it was demonstrated that certain miRs were associated to lymph node status and survival rates. However, these miRs were all extracted from paraffin-preserved tissues of patients having had a hysterectomy for EC and were not studied in their extracellular compartments. In our systematic literature review, a total of 25 circulating miRs have been analyzed for their clinical implications. Three miRs were significantly different for the histologic grading (miR-9, miR-21, miR-142-3p). MiR-9 is a tumor suppressor in numerous cancers, such as colorectal cancer, acute myeloid leukemia, and nasopharyngeal carcinoma [42,43,44]. In colorectal cancer [42] and acute myeloid leukemia [43], miR-9 targets and represses C-X-C Motif Chemokine Receptor 4 (CXCR4), which influences cell proliferation and the epithelial–mesenchymal transition (EMT). In nasopharyngeal carcinoma, exosomal miR-9 has been shown to target the MDK and PDK/AKT signaling pathway and inhibit the formation and migration of endothelial tubes. In this study [44], the expression of exosomal miR-9 was correlated to overall survival. MiR-142-3p is implicated in the pathogenesis of many gynecological cancers. This miR targets Sirtuin 1 (SIRT1) and represses the proliferation and chemoresistance in ovarian cancer [45] and targets the Wiskott–Aldrich syndrome-like protein (WASL), Integrin Alpha V, and other cytoskeletal components. Also, it inhibits invasiveness in breast cancer [46]. MiR-142-3p has also been linked to endometriosis. In a study of Ma et al. [47], miR-142-3p regulated VEGFA expression and targeted KLF9 in vitro and attenuated ectopic endometriotic lesions in vivo. The molecular functions of miR-9 and miR-142-3p in EC remain to be understood. However, the physiopathology of miR-21 has been described in EC. The overexpression of miR-21 significantly reduces SRY-box 17 (SOX17) protein expression and generates the epithelial–mesenchymal transition (EMT) in EC cells [48]. Four miRs were significantly different concerning the different FIGO stages of EC (miR-21, miR-29b, miR-99a, miR-449a). In a study of Jing et al.[49], it has been shown that miR-499 targets the 3′ UTR region of VAV3 and that its circulation in exosomes inhibits EC cell proliferation, tube formation, and angiogenesis. The mechanism of miR-99 in EC has not been described in the literature. It has been suggested that miR-99 reduces DNA damage repair by targeting SNF2H [50]. In this systematic review of literature, miR-29b was significantly different for many clinical aspects, such as tumor size, lymph node status, and survival. This miR has been described as having many different implications in EC [51]. By regulating the expression of BAX and Bcl-2, it increased the sensitivity of EC cells to cisplatin and therefore augmented secondary apoptosis. MiR-29b also limits proliferation and reduces the migration and invasion of EC cells. Additionally, miR-29 binds to the 3′ UTR of PTEN and changes its expression level.

MiRs seem to be effective markers in the diagnosis of EC and some may even be linked to certain prognosis factors. However, an interpretation of the results of studies concerning miR expression should be done with great caution. In this systematic review, some authors bring contradictory results concerning the expression of certain miRs and their association to prognostic factors in EC. The small number of patients in these studies could partly be responsible, except the study of Wang et al. [15], which included 356 EC patients and 155 healthy patients—the maximum number of patients included was 50 for EC and healthy patients, respectively. When analyzing subgroups (FIGO stage, histological grading…), the number of patients was even lower. For example, in the study of Fan et al. [17], miR-204-5p was significantly upregulated in G1 and G2 EC patients compared to healthy patients, but this difference of expression w[s not statistically significant for G3. This was probably related to the small number of patients included in this group: 12 EC patients with Grade 3 compared to 36 with Grade 1 and 44 with Grade 2.

This low number of patients included influences the statistical power of the results and makes them difficult to interpretate. Multicentric studies can compensate for this lack of power and can help to harmonize our practices. The discrepancy of some results could also be related to the regulatory mechanisms of miRs that we have yet to understand.

Another element that should bring caution to the interpretation of the results of these studies is the presence of missing information. Many studies analyzed the diagnostic performance of certain miRs, but there is often information lacking regarding the diagnostic cutoff value. In this systematic review, out of the 13 studies that analyzed the diagnostic performance of miRs in EC, only 6 gave information concerning the cutoff value. None of the authors provided information on cutoff values that could be associated with the histopathological aspects of EC compared to healthy patients.

In addition to these limiting factors, another difficulty in miR analysis is the variety of evaluation tools utilized. In the studies cited, the criteria and methods of analysis varied widely (different histological grading groups, FIGO stages groups, samples, kits, conservation techniques), making comparison between them difficult. For example, the origin of the samples differed as miRs were extracted from serum in seven articles, plasma in six articles, and venous blood in one article. The information about the reproducibility of serum or plasma analysis is inconsistent. The study by Mitchell et al. suggests that the quantification of miRs in plasma and serum are correlated and that the analysis of miRs from plasma or serum are therefore reliable sources of sampling [52]. However, other studies have shown that miR expression is different according to the origin of the sample (plasma or serum). It has been suggested that miR expression is more elevated in serum than in plasma samples. This difference in expression could be explained by the association of certain miRs to platelets, which are responsible of modifying the expression levels of miRs in blood after the coagulation process [53]. In a study regarding the adenocarcinoma of the esophagus, miRs extracted from serum could outperform miRs extracted from plasma as the format of the miRs differed [54] In the serum, vesicle-associated miRs were expressed at higher levels than in the plasma, whereas the plasma mostly contained non-vesicle-associated miRs. The expression and function of miRs also appeared to vary between arterial and venous blood samples [55]. Variations have also been observed according to the different kits that were used (Table 5). In this systematic review of literature, there was a large variation of kits that were used—a total of ten different kits were employed. It has been implied that, depending on the kit used, the quality and quantity assessment of miR varied and that the miRNeasy Serum/Plasma kit surpasses other kits [56]. The variety of kits used is a major limitation in our systematic review of literature as it makes the studies difficult to compare. There is an urgent need to establish recommendations for miR analysis in order to harmonize our practices.

For all these reasons, a general synthesis of the results is difficult to achieve. In addition, many of the studies analyzed did not consider the possible factors that may influence miR expression. Many factors can influence the expression levels of certain miRs and one of them could be menopausal status. In this systematic review of literature, only three studies evaluated the variation of miR expression with menopausal status. MiR-142-3p and miR-146-5p had higher expression levels in patients without menopause compared to menopaused patients [19]. MiR-150-p was over-expressed in menopaused patients compared to patients without menopause (*p* = 0.005) [20]. MiR-29b expression did not significantly vary according to menopausal status (*p* = 0.195) [15]. When comparing to other studies, the variation of expression regarding menopausal status was also observed for miR-182 in patients with locally advanced triple negative breast cancer. The variation of miR-182 expression was significantly different between patients without menopause and menopaused patients (*p* = 0.009), with an absence of the over-expression of miR-182 in menopaused patients [57]. Another influencing factor could be the age of the patients. In this systematic review of literature, only one study evaluated the variation of miR expression according to the age of the patient [15]. In this study, miR-29b expression did not significantly change between patients <50 and patients >50 years old (*p* = 0.172). A study of Huan et al. [58] evaluated the variation of expression of miRs in 5221 patients and found that 127 miR were differently expressed according to age. In this systematic review of literature, no studies analyzed the variation of miR expression according to body mass index (BMI). BMI can influence miR expression as shown in the study of Hijmans et al. [59]. In this study, miR-34a expression was significantly higher in obese patients compared to normal-weight and overweight patients, whereas miR-126, miR-146a, and miR-150 expression were significantly lower in obese and overweight patients compared to normal-weight patients. MiR-181b expression did not significantly vary between these different weight groups. The lack of analysis of miR expression according to these different clinical features is an important limitation in the comparison of these studies. MiR expression can vary among the different studies according to the type of characteristics of the patients included. These influencing factors are essential to take into account in order to harmonize our studies.

A comparison of the expression of circulating miRs between different studies is therefore very complicated and the heterogeneity of the results is explained by numerous factors. When comparing the 33 miRs of this systematic review to a miR cancer database, such as the Database of Differentially Expressed miRNAs in Human Cancers (dbDEMC), a variety of results were observed. The expression levels varied for 14 miRs: miR-9, miR-30a-5p, and miR-301b were upregulated according to dbDEMC, whereas they were downregulated in our studies; and miR-141, miR-142-3p, miR-143-3p, miR-150-5p, miR-186, miR-195-5p, miR-199b, miR-200a, miR-203, miR-205, and miR-223 were downregulated according to dbDEMC, whereas they were upregulated in our systematic review of literature. Four miRs had not been mentioned in dbDEMC: miR-449, miR-887-5p, miR-1228, and miR-1290 (Table 6). This difference of expression can be explained by the fact that we studied circulating miRs in our systematic review of literature, whereas the miRs mentioned in dbDEMC were essentially extracted from primary tumors.

Less invasive tests are being studied for the detection of miRs and some studies have shown their presence in other fluid samples, such as saliva and urine [60,61]. In a study of O’Flynn et al. [62], urine and vaginal samples were analyzed for their diagnostic accuracy in EC. Urine cytology had a sensitivity and a specificity of 72% and 94.9% with a diagnostic accuracy of 83.2%, whereas vaginal cytology had a sensitivity and specificity of 89.6% and 88.7% with a diagnostic accuracy of 89.2%. Combining vaginal cytology to urine cytology allowed for the amelioration of the diagnostic accuracy in EC of up to 90.3%. It is important to verify the validity and reproducibility of these less invasive tests, as they may be interesting alternatives to blood samples if they are proven to be accurate in the detection of EC and are associated with prognostic factors.

MiRs have been proven to be stable markers and resistant to RNase activity and environmental factors. In a study by Mitchell et al. [52], miR levels did not change when modifying the temperature of the samples (room temperature incubation for up to 24 h or freeze–thawing for up to 8 cycles). It was also shown that endogenous miRs were resistant to RNase activity, whereas exogenous miRs were rapidly degraded when introduced. This stability is due to the fact that miRs circulate within exosomes, which protect them from deterioration. Exosomes are nanometric (30–200 nm) vesicles that are released by a cell and contain many elements, such as lipids, proteins, glycoconjugates, and nucleic acids. They play an essential role in intercellular communication and have now been proven to be implicated in pathological processes [63]. Among the many elements that they contain, they also transport different miRs and are therefore implicated in different cancer progression pathways [64]. In breast cancer derived exosomes, precursor miRs were found with their processing complex (Dicer, AGO), which suggests that miRs could travel extracellularly in their precursor state and be produced directly in the exosomes [65]. Another theory is that the majority of miRs in plasma are connected directly to the AGO protein and that it is this complex that explains its resistance to RNase activity and environmental factors [66]. Further studies are needed to understand the mechanism of circulation miRs in the extracellular compartments.

## 5. Conclusions

In this systematic review of literature, a total of 33 circulating miRs and 6 different panels of circulating miRs have been described for their diagnostic performance in EC diagnosis. A total of seven circulating miRs and one panel of circulating miRs have been associated with clinical and prognostic factors in EC. This minimally invasive analysis of miRs may help to better guide the management of EC patients. However, further studies are necessary to better understand their function, mechanism of pathogenesis, and extracellular circulation.

## Figures and Tables

**Figure 1 cells-11-01836-f001:**
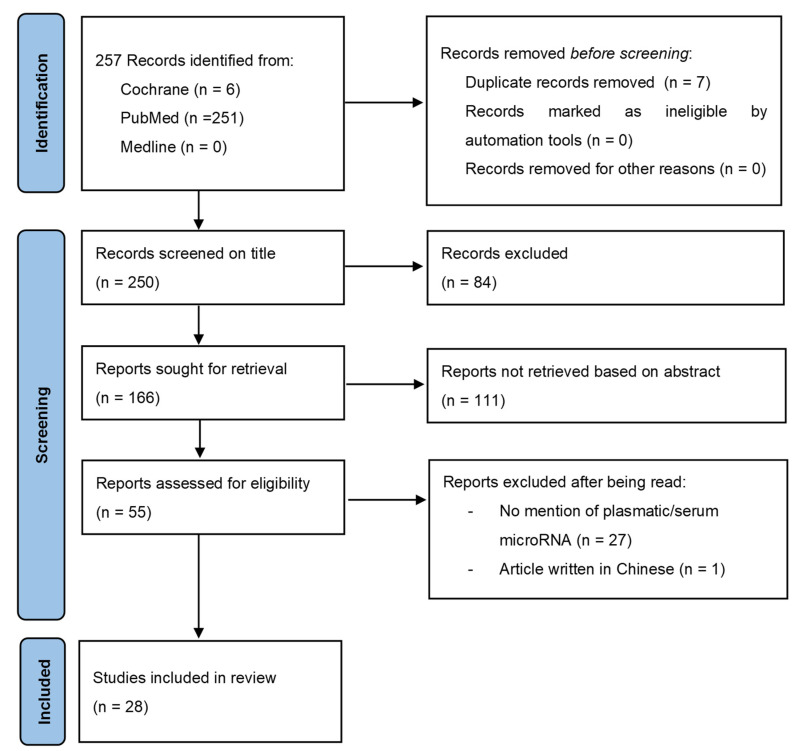
PRISMA flow diagram.

**Figure 2 cells-11-01836-f002:**
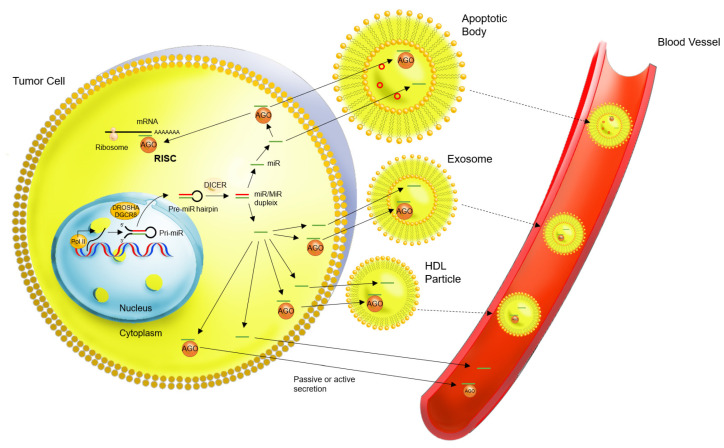
Molecular function of circulating miRs in EC. miR: microRNA, PolII: RNA polymerase II, pri-miR: primary transcript of microRNA, pre-miR: precursor of microRNA, AGO: Argonaute, mRNA: RNA messenger, RISC: RNA induced silencing complex.

**Table 1 cells-11-01836-t001:** List of articles concerning circulating miRs in endometrial cancer.

Authors	Date	Publication	Name of article
**Original Article**			
Torres, A. et al. [7]	2012	BMC Cancer	Deregulation of miR-100, miR-99a and miR-199b in tissues and plasma coexists with increased expression of mTOR kinase in endometrioid endometrial carcinoma
Jia, W. et al. [8]	2013	Oncol. Lett.	Identification of four serum microRNAs from a genome-wide serum microRNA expression profile as potential non- invasive biomarkers for endometrioid endometrial cancer
Torres, A. et al. [9]	2013	Int. J. Cancer.	Diagnostic and prognostic significance of miRNA signatures in tissues and plasma of endometrioid endometrial carcinoma patients
Wang, L. et al. [10]	2014	PLoS ONE	Circulating microRNAs as a fingerprint for endometrial endometrioid adenocarcinoma
Tsukamoto, O. et al. [11]	2014	Gynecol. Oncol.	Identification of endometrioid endometrial carcinoma-associated microRNAs in tissue and plasma
Jiang, Y. et al. [12]	2016	Int. J. Gynecol. Cancer	Changes in the Expression of Serum MiR-887-5p in Patients With Endometrial Cancer
Montagnana, M. et al. [13]	2017	Int. J. Gynecol. Cancer	Aberrant MicroRNA Expression in Patients With Endometrial Cancer
Benati, M. et al. [14]	2017	Clin. Lab.	Evaluation of mir-203 Expression Levels and DNA Promoter Methylation Status in Serum of Patients with Endometrial Cancer
Wang, H. et al. [15]	2018	Future Oncol.	Expression and prognostic value of miRNA-29b in peripheral blood for endometrial cancer
Ritter, A. et al. [16]	2020	Cancer Biomark.	Discovery of potential serum and urine-based microRNA as minimally-invasive biomarkers for breast and gynecological cancer
Fan, X. et al. [17]	2021	Biosci. Rep.	MicroRNA expression profile in serum reveals novel diagnostic biomarkers for endometrial cancer
Zhou, L. et al. [18]	2021	Mol. Cancer	Plasma-derived exosomal miR-15a-5p as a promising diagnostic biomarker for early detection of endometrial carcinoma
Fan, X. et al. [19]	2021	Cancer Biomark.	Three plasma-based microRNAs as potent diagnostic biomarkers for endometrial cancer
Ghazala, R.A. et al. [20]	2021	Mol. Biol. Rep.	Circulating miRNA 27a and miRNA150-5p; a noninvasive approach to endometrial carcinoma
**Original Article with Meta-Analysis**			
Gao, Y. et al. [21]	2016	Oncotarget	Diagnostic value of circulating miR-21: An update meta-analysis in various cancers and validation in endometrial cancer
**Meta-Analysis**			
Liu, X. et al. [22]	2016	Am. J. Ther.	MicroRNA-200 Family Profile: A Promising Ancillary Tool for Accurate Cancer Diagnosis
Wang, F. et al. [23]	2019	Clin. Lab.	The Significance Role of microRNA-200c as a Prognostic Factor in Various Human Solid Malignant Neoplasms: A Meta-Analysis
**Systematic Review**			
Rižner, T.L. [24]	2016	Expert Rev. Mol. Diagn.	Discovery of biomarkers for endometrial cancer: current status and prospects
Donkers, H. et al. [25]	2020	Oncotarget	Diagnostic value of microRNA panel in endometrial cancer: A systematic review
Klicka, K. et al. [26]	2021	Cancers	The Role of miRNAs in the Regulation of Endometrial Cancer Invasiveness and Metastasis—A Systematic Review
**Review**			
Zhao, Y.N. et al. [27]	2014	Exp. Hematol. Oncol.	Circulating MicroRNAs in gynecological malignancies: from detection to prediction
Yanokura, M. et al. [28]	2015	EXCLI J.	MicroRNAS in endometrial cancer: recent advances and potential clinical applications
Zavesky, L. et al. [29]	2015	Neoplasma	New perspectives in diagnosis of gynecological cancers: Emerging role of circulating microRNAs as novel biomarkers
Kanekura, K. et al. [30]	2016	J. Obstet. Gynaecol. Res.	MicroRNA and gynecologic cancers
Muinelo-Romay, L. et al. [31]	2018	Int. J. Mol. Sci.	Liquid Biopsy in Endometrial Cancer: New Opportunities for Personalized Oncology
Song, Q. et al. [32]	2019	J. Oncol.	Role of miR-221/222 in Tumor Development and the Underlying Mechanism
De Bruyn, C. et al. [33]	2020	Curr. Oncol. Rep.	Circulating Transcripts and Biomarkers in Uterine Tumors: Is There a Predictive Role?
Openshaw, M.R. et al. [34]	2020	Front. Digit Health	Non-invasive Technology Advances in Cancer-A Review of the Advances in the Liquid Biopsy for Endometrial and Ovarian Cancer

miRs: microRNAs.

**Table 2 cells-11-01836-t002:** Expression and diagnostic performance of circulating miRs in EC patients compared to healthy patients.

Name of miR		Sample	EC Patients n	Healthy Patients (Without EC) n	Circulating miR Variation in EC Vs. Healthy Patients	AUC (95% CI/p)		Cut-Off Value	Se	Spe
						Training Phase (TP)	Validation Phase (VP)			
**Individual miR**										
miR-9 [9]		plasma	34	14	Down	-	0.768 (0.622–0.879)	2.6	88	71
miR-15b [10]		plasma	Screening phase: 9 Validation phase: 31	Screening phase: 20 Validation phase: 33	Up	-	0.768 (0.653–0.882)	-	74.2	69.7
miR-20b-5p [17]		serum	Screening phase: 2 Testing phase: 21 Validation phase: 41 External validation: 30	Screening phase: 1 Testing phase: 24 Validation phase: 48 External validation: 30	Up	0.756 (0.689-0.823) ^†^		-	-	-
miR-21	[21]	serum	50	50 (50 *)	Up (Up *)	-	0.831 (0.746–0.916) 0.710 * (0.608–0.813) *	2.937 (3.457 *)	70 (64 *)	92 (76 *)
	[11]	serum	12	12	Down	-	0.757 (0.561–0.953)	-	-	-
miR-27a	[10]	plasma	Screening phase: 9 Validation phase: 31	Screening phase: 20 Validation phase: 33	Up	-	0.813 (0.699–0.927)	-	77.4	81.8
	[20]	serum	36	36	Up	-	1.000 (<0.001)	0.2872	100	100
miR-29b	Located EC [15]	Venous blood **	356	155 (149 *)	Down	-	0.976 (0.951–1.000)	0.940	96.1	97.9
	Metastatic EC [15]	Venous blood **	356	155 (149 *)	Down	-	0.974 (0.949–0.999)	0.917	96.7	95
miR-30a-5p [11]		Plasma	12	12	Down	-	0.813 (0.638–0.987)	-	-	-
miR-92a [9]		Plasma	34	14	Up	-	0.794 (0.651–0.898)	1.6	61	93
miR-99a [7]		Plasma	34	14	Up	-	0.810 (0.669–0.909)	1.23	76	79
miR-100 [7]		Plasma	34	14	Up	-	0.740 (0.592–0.857)	1.5	64	79
miR-135b [11]		Plasma	12	12	Up	-	0.972 (0.913–1.000)	-	-	-
miR-141 [9]		plasma	34	14	Up	-	0.766 (0.620–0.877)	2.5	58	93
miR-142-3p [19]		Plasma	Screening phase: 2 Testing phase: 22 Validation phase: 44 External validation: 27	Screening phase: 1 Testing phase: 22 Validation phase: 34 External validation: 23	Up	0.689 (0.611–0.767) ^†^		-	-	-
miR-143-3p [17]		serum	Screening phase: 2 Testing phase: 21 Validation phase: 41 External validation: 30	Screening phase: 1 Testing phase: 24 Validation phase: 48 External validation: 30	Up	0.677 (0.602–0.751) ^†^		-	-	-
miR-146a-5p [19]		Plasma	Screening phase: 2 Testing phase: 22 Validation phase: 44 External validation: 27	Screening phase: 1 Testing phase: 22 Validation phase: 34 External validation: 23	Up	0.694 (0.616-0.772) ^†^		-	-	-
miR-150-5p [20]		Serum	36	36	Up	-	0.982 (<0.001)	1.02	88.89	100
miR-151a-5p [19]		Plasma	Screening phase: 2 Testing phase: 22 Validation phase: 44 External validation: 27	Screening phase: 1 Testing phase: 22 Validation phase: 34 External validation: 23	Up	0.680 (0.601–0.759) ^†^		-	-	-
miR-186	[13] ***	Serum	46	28	Up	-	0.7000 (=0.004)	-	-	-
	[8]	Serum	Screening phase: 7 Validation phase: 26	Screening phase: 20 Validation phase: 22	Up	-	0.865 (0.755–0.974)	-	-	-
miR-195-5p [17]		serum	Screening phase: 2 Testing phase: 21 Validation phase: 41 External validation: 30	Screening phase: 1 Testing phase: 24 Validation phase: 48 External validation: 30	Up	0.669 (0.593–0.745) ^†^		-	-	-
miR-199b [7]		Plasma	34	14	Up	-	0.786 (0.642–0.892)	2.48	79	71
miR-200a [9]		Plasma	34	14	Up	-	0.792 (0.649–0.897)	2.2	67	93
miR-203	[14]	Serum	45	30	Up	-	0.710 (0.590–0.830)	-	-	-
	[9]	Plasma	34	14	Up	-	0.766 (0.620–0.877)	3.3	64	93
miR-204	[13] ***	Serum	46	28	Down	-	1.000 (<0.0001)	-	100	100
	[8]	Serum	Screening phase: 7 Validation phase: 26	Screening phase: 20 Validation phase: 22	Up	-	0.740 (0.594–0.885)	-	-	-
	[17]	serum	Screening phase: 2 Testing phase: 21 Validation phase: 41 External validation: 30	Screening phase: 1 Testing phase: 24 Validation phase: 48 External validation: 30	Up	0.668 (0.592–0.743) ^†^		-	-	-
miR-205 [11]		Plasma	12	12	Up	-	1.000 (1.000–1.000)	-	-	-
miR-222	[13] ***	Serum	46	28	Up	-	0.720 (=0.002)	-	-	-
	[8]	Serum	Screening phase: 7 Validation phase: 26	Screening phase: 20 Validation phase: 22	Up	-	0.837 (0.726–0.948)	-	-	-
miR-223	[13] ***	Serum	46	28	Up	-	0.880 (<0.0001)	-	-	-
	[8]	Serum	Screening phase: 7 Validation phase: 26	Screening phase: 20 Validation phase: 22	Up	-	0.727 (0.577–0.877)	-	-	-
	[10]	Plasma	Screening phase: 9 Validation phase: 31	Screening phase: 20 Validation phase: 33	Up	-	0.768 (0.651–0.885)	-	64.5	81.8
miR-301b [9]		Plasma	34	14	Down	-	0.660 (0.507–0.792)	2.3	55	86
miR-423-3p [17]		serum	Screening phase: 2 Testing phase: 21 Validation phase: 41 External validation: 30	Screening phase: 1 Testing phase: 24 Validation phase: 48 External validation: 30	Up	0.689 (0.611–0.767) ^†^		-	-	-
miR-449 [9]		Plasma	34	14	Up	-	0.879 (0.750–0.956)	5.5	91	86
miR-484 [17]		serum	Screening phase: 2 Testing phase: 21 Validation phase: 41 External validation: 30	Screening phase: 1 Testing phase: 24 Validation phase: 48 External validation: 30	Up	0.644 (0.566–0.722) ^†^		-	-	-
miR-887-5p [12]		Serum	Screening phase: 50 Validation phase: 20	Screening phase: 50 Validation phase: 20	Up	-	0.729 (0.563–0.892)	-	60	95
miR-1228 [9]		Plasma	34	14	Up	-	0.890 (0.764–0.962)	4	73	100
miR-1290 [9]		Plasma	34	14	Up	-	0.773 (0.627–0.882)	1.9	76	86
**Association of miR**										
miR-222, miR-223, miR-186, miR-204 [8]		Serum	Screening phase: 7 Validation phase: 26	Screening phase: 20 Validation phase: 22		-	0.927 (0.845–1.000)	-	91.7	87.5
miR-142-3p, miR-146a-5p, miR-151a-5p [19] ****		Plasma	Screening phase: 2 Testing phase: 22 Validation phase: 44 External validation: 27	Screening phase: 1 Testing phase: 22 Validation phase: 34 External validation: 23		0.729 (0.580–0.879)	0.751 (0.645–0.858) ^±^	0.528 ^†^	62 ^†^	64.5^†^
miR-143-3p, miR-195-5p, miR-20b-5p, miR-204-5p, miR-423-3p, miR-484 [17] ****		serum	Screening phase: 2 Testing phase: 21 Validation phase: 41 External validation: 30	Screening phase: 1 Testing phase: 24 Validation phase: 48 External validation: 30		0.748 (0.599–0.897)	0.833 (0.745–0.921) ^‡^	-	TP: 83.3 VP: 77.1	TP: 66.7 VP: 82.9
miR-9/miR-92a [9]		Plasma	34	14		-	0.909 (0.789–0.973)	0.89	73	100
miR-9/miR-1229 [9]		Plasma	34	14		-	0.913 (0.794–0.976)	0.83	79	100
miR-99a/miR-199b [7]		Plasma	34	14		-	0.903 (0.780–0.970)	0.73	88	93
**Association miR and Other Markers**										
miR-27a and CA 125 [10]		plasma	Screening phase: 9 Validation phase: 31	Screening phase: 20 Validation phase: 33		-	0.894 (0.807–0.980)	-	77.4	97

miR: microRNA; EC: endometrial cancer; n: number of patients; AUC: area under the curve; Se: sensitivity; Spe: specificity; up: overexpression; down: under expression; -: not mentioned in the article; *: comparison of EC patients with patients that have benign uterine lesions (polyps, myomas); **: authors did not specify origin of miRs (serum or plasma); ***: remained significant when comparison of FIGO stage I EC patients with healthy patients without EC; ****: remained significant when comparison of EC patients with FIGO stage I, stage II-IV, histological grade 1, histological grade 2, and histological grade 3 with healthy patients without EC: ^†^: combined data of training, testing, and external stages; ±: external validation stage: AUC = 0.789 (95% CI: 0.664–0.914) and combined data of training, testing, and external validation stages: AUC = 0.716 (95% CI: 0.640–0.793); ^‡^: external validation phase: 0.967 (95% CI: 0.928–1.000), Se 83.3%, Spe 100% and combined data of training and testing stages: AUC = 0.775 (95% CI: 0.710–0.840), Se78.4%, Spe 63%.

**Table 3 cells-11-01836-t003:** Clinical and prognostic characteristics of circulating miRs in EC patients compared to healthy patients without EC.

Clinical and Prognostic Characteristic	Upregulated	Downregulated	NS
**Histological Grade**			
G1	[9]: miR-92a, miR-141, miR-200a, miR-203, miR-449a, miR-1228, miR-1290 [17]: miR-20b-5p, miR-143-3p, miR-195-5p, miR-204-5p, miR-423-3p, miR-484	[9]: miR-9	[9]: miR-301b
G2	[17]: miR-20b-5p, miR-143-3p, miR-195-5p, miR-204-5p, miR-484		[17]: miR-423-3p
G3	[17]: miR-20b-5p, miR-143-3p, miR-195-5p, miR-423-3p, miR-484		[17]: miR-204-5p
G2–G3	[9]: miR-92a, miR-141, miR-200a, miR-449a, miR-1228, miR-1290	[9]: miR-9	[9]: miR-203, miR-301b
**FIGO Stages**			
I	[17]: miR-20b-5p, miR-143-3p, miR-195-5p, miR-204-5p, miR-423-3p, miR-484 [13]: miR-186 *, miR-222 *, miR-223 *	[13]: miR-204 *	
II–IV	[17]: miR-20b-5p, miR-143-3p, miR-195-5p, miR-423-3p, miR-484		[17]: miR-204-5p
IA	[9]: miR-92a, miR-141, miR-203, miR-449a, miR-1228, miR-1290 [7]: miR-99a, miR-199b	[9]: miR-9	[9]: miR-200a, miR-301b [7]: miR-100
>IA	[9]: miR-92a, miR-141, miR-200a, miR-203, miR-449a, miR-1228, miR-1290 [7]: miR-99a, miR-100, miR-199b	[9]: miR-9	[9]: miR-301b

miR: microRNA; EC: endometrial cancer; NS: statistically not significant; G1: histological grade 1; G2: histological grade 2; G3: histological grade 3; FIGO: International Federation of Gynecology and Obstetrics. *: miR-186: AUC 0.73 (*p* = 0.002), miR-222: AUC 0.71 (*p* = 0.006), miR-223: AUC 0.85 (*p* = 0.0001), miR-204: AUC 1 (*p* < 0.0001).

**Table 4 cells-11-01836-t004:** Comparison of clinical and prognostic characteristics of circulating miRs between EC patients.

miR		Histological Type	Histological Grade	Primitive Tumor Size	Myometrial Invasion	FIGO Stage	LNM	LVSI	Distant Metastasis	Average Survival Rate
miR-9 [9]		-	X	-	NS	NS	-	-	-	-
miR-21 [11]		-	X	-	-	X	-	-	-	-
miR-27a [20]		X	NS	-	NS	NS	NS	NS	-	-
miR-29b [15]		NS	NS	X	NS	X	X	-	-	X
miR-30a-3[11]		-	NS	-	-	NS	-	-	-	-
miR-92a [9]		-	NS	-	NS	NS	-	-	-	-
miR-99a [7]		-	NS	-	NS	X	-	-	-	-
miR-100 [7]		-	NS	-	NS	NS	-	-	-	-
miR-135b [11]		-	NS	-	-	NS	-	-	-	-
miR-141 [9]		-	NS	-	NS	NS	-	-	-	-
miR-142-3p [19]		-	X	-	-	NS	-	-	-	-
miR-146a-5p [19]		-	-	-	-	NS	-	-	-	-
miR-150-5p [20]		NS	NS	-	NS	NS	NS	NS	-	-
miR-151-5p [19]		-	-	-	-	NS	-	-	-	-
miR-186 [13]		-	NS	-	-	-	-	-	-	-
miR-199b [7]		-	NS	-	NS	NS	-	-	-	-
miR-200a [9]		-	NS	-	NS	NS	-	-	-	-
miR-203	[14]	-	NS	-	-	NS	-	-	-	-
	[9]	-	NS	-	NS	NS	-	-	-	-
miR-205 [11]		-	NS	-	-	NS	-	-	-	-
miR-222 [13]		-	NS	-	-	-	-	-	-	-
miR-223 [13]		-	NS	-	-	-	-	-	-	-
miR-301b [9]		-	NS	-	NS	NS	-	-	-	-
miR-449a [9]		-	NS	-	NS	X	-	-	-	-
miR-1228 [9]		-	NS	-	NS	NS	-	-	-	-
miR-1290 [9]		-	NS	-	NS	NS	-	-	-	-
Panel miR-200b/miR-200c/miR-203/miR-449a [9]		-	-	-	X	-	-	-	-	-

miR: microRNA; FIGO: International Federation of Gynecology and Obstetrics; LNM: lymph node metastasis; LVSI: lymphovascular space involvement; -: not mentioned; X: statistically significant; NS: statistically not significant.

**Table 5 cells-11-01836-t005:** Conservation, extraction, and analysis of the miRs.

Ref	Type of Sample	Conservation	Extraction	Micro-Array	qT-PCR
+6969.68/7 [7]	Plasma	−80 °C	mirVana Paris Kit (Ambion)		Precision nanoScript Reverse Transcription kit (Primer Design)
[8]	Serum	−70 °C	TRIzol reagent (Invitrogen)	TaqMan microRNA RT kit and Megaplex RT primers (Invitrogen)	AMV reverse transcriptase (Takara Dalian, Liaoning, China) and the stem-loop RT primer (Applied Biosystems)
[9]	Plasma	−80 °C	mirVana Paris Kit (Ambion)		TaqMan MicroRNA Reverse Transcription Kit and specific primers (Applied Biosystems)
[10]	Plasma	−80 °C	miRcute miRNA Isolation Kit	Sharpvue 26 Universal qPCR Master Mix High Rox kit (Biovue, Shanghai, China) and Sharpvue Human miRNA Primer Array kit (Biovue, Shanghai, China)	Sharpvue miRNA First Strand Kit (Biovue, Shanghai, China)
[11]	Plasma	−80 °C	mirVana Paris Kit (Ambion)		TaqMan MicroRNA Assays (Applied Biosystems)
[12]	Serum	−80 °C	mirVana Paris Kit (Ambion)	Solexa sequencing	PrimeScript RT Reagent Kit et SYBR Premix Ex Taq Kit
[21]	Serum	-	miRNeasy Serum/Plasma Kit (Qiagen)		miScript II RT Kit and miScript SYBR Green PCR Kit (Qiagen)
[13]	Serum	−80 °C	mirVana Paris Kit (Ambion)		TaqMan MicroRNA Assay
[14]	Serum	−80 °C	mirVana PARIS Kit (Ambion)		TaqMan Advanced miRNA cDNA synthesis Kit
[15]	Venous blood	-	RNA extraction kit (Shanghai LifeFeng Biotech Co.)		RNA reverse transcription kit (ThermoFisher Scientific)
[16]	Serum	−20 °C	Norgen Total RNA Purification Kit	Human miRNA microarray chip analysis (Agilent-070156 Human)	Reaction mix
[17]	Serum	−80 °C	mirVana Paris Kit (Ambion)	Exiqon miRCURY-Ready-to-Use PCR-Human-panel- I+II-V1.M	Bulge-Loop™ miRNA qRT-PCR primer set
[18]	Plasma	−80 °C	miRNeasy Micro Kit (QIAgen)	QIAquick PCR Purification Kit (QIAgen)	KAPA Library Quantification Kit (KAPA Biosystems)
[19]	Plasma	−80 °C	mirVana Paris Kit (Ambion)	Exiqon miRCURY Ready-to-Use PCR-Human-panel- I+II-V1.M	Bulge-LoopTM miRNA qRT-PCR Primer Set
[20]	Serum	-	miRNeasy Micro Kit (QIAgen)		TaqMan MicroRNA Reverse Transcription Kit

Ref: reference of the article; miR: microRNA; -: not mentioned.

**Table 6 cells-11-01836-t006:** Comparison of the variation of expression in EC patients compared to healthy patients of the 33 miRs in our systematic review with dbDEMC.

Name of miR		Circulating miR Variation in EC Vs. Healthy Patients	miRNA-Cancer Data Base (dbDEMC)
miR-9 [9]		Down	Up
miR-15b [10]		Up	Up
miR-20b-5p [17]		Up	Up
miR-21	[21]	Up (Up *)	Up
	[11]	Down	
miR-27a	[10]	Up	Up
	[20]	Up	
miR-29b	Located EC [15]	Down	Down
	Metastatic EC [15]	Down	
miR-30a-5p [11]		Down	Up
miR-92a [9]		Up	Up
miR-99a [7]		Up	Up
miR-100 [7]		Up	Up
miR-135b [11]		Up	Up
miR-141 [9]		Up	Down
miR-142-3p [19]		Up	Down
miR-143-3p [17]		Up	Down
miR-146a-5p [19]		Up	Up
miR-150-5p [20]		Up	Down
miR-151a-5p [19]		Up	Up
miR-186	[13]	Up	Down
	[8]	Up	
miR-195-5p [17]		Up	Down
miR-199b [7]		Up	Down
miR-200a [9]		Up	Down
miR-203	[14]	Up	Down
	[9]	Up	
miR-204	[13]	Down	Down
	[8]	Up	
	[17]	Up	
miR-205 [11]		Up	Down
miR-222	[13]	Up	Up
	[8]	Up	
miR-223	[13]	Up	Down
	[8]	Up	
	[10]	Up	
miR-301b [9]		Down	Up
miR-423-3p [17]		Up	Up
miR-449 [9]		Up	-
miR-484 [17]		Up	Up
miR-887-5p [12]		Up	-
miR-1228 [9]		Up	-
miR-1290 [9]		Up	-

miR: microRNA; EC: endometrial cancer; dbDEMC: Database of Differentially Expressed miRNAs in Human Cancers; -: not mentioned.

## Data Availability

Not applicable.

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
