# Peer review of "Clinical Value and Molecular Function of Circulating MicroRNAs in Endometrial Cancer Regulation: A Systematic Review"

_cells, 2022, doi:10.3390/cells11111836_

Round 1
Reviewer 1 Report
The systemic review article, entitled " Clinical value and molecular function of circulating microRNAs in endometrial cancer regulation: a systematic review" is interesting and contains updated data. The article describes different microRNAs in serum or plasma of endometrial cancer patients and the association with clinical and prognostic characteristics. There are 33 circulating miRNA and 6 panels show significant expression differences in endometrial cancer patients. Seven circulating miRNAs and 1 panel are found in association with clinical and prognostic characteristics. The authors also discussed the issues regarding the miRNA extraction methods and the differences of the miRNAs in the serum and plasma. This article could help the readers understand which miRNAs could be used as potential biomarkers for diagnosis and management of endometrial cancer.
I wound recommend its publication after major revision.
- All words in Figure 1 should be shown more clearly and edited.
- miR-204-5p was up-regulated in the patients with G2, but statistically not significant in G3 and G2-G3. Could you please explain the results? miR-204 was up-regulated in EC patients (Ref. 14 and 18) but was found to be down-regulated (Ref 8), compared to healthy subjects (Table 2). Contrarily, miR-204 was down-regulated in stage I EC (Ref. 14), but was not shown to be upregulated in stage II-IV (Table 3). Could authors explain why its expression differs in the same reference?
- Authors reviewed the evaluations the diagnostic performance of circulating miRs in various grading system in EC, such as (Table 3). Does any trend in the concentrations of circulating miRs exist regarding the extent of the histological grade, myometrial invasion and FIGO stage, when compared to the health subjects?
- Line 290. Regarding the prognostic role of miR-9, it was overexpressed in G2 and G3 compared to G1 EC patients (Ref 9). Regarding the diagnostic role of miR-9 (In Table 3), it was down-regulated in G1 and G2-G3 (Ref. 9), compared to healthy patients. As an example, authors may explain the discrepancies between the demonstrated results.
- Line 311. The reference should be inserted in this sentence.
- Line 318. The plasma concentration of miR-21 was significantly higher in EC patients with FIGO stage 1A.
In Table 3, there is no miR-21 showing “up” in FIGO stage 1A.
- It is difficult to read the words in Figure 2.
- Line 383-384. In the context of EC, three miRs (miR-142-3p, miR-146a-5p, and miR-151a-5p) were found to be over-expressed in the plasma of EC patients compared to healthy patients. miR-142-3p and miR-146a-5p are not shown in the Table 3. Could you please confirm and illustrate?
- Authors should discuss more the inconsistences between studies and the limitations of this review work in Discussion section.
- English Editing to improve the article should be conducted.
Reviewer 2 Report
Methods:
- Bias assessment? why not applicable? Clearly explain in methods section
Results
- “best diagnostic performance” (page 9 lane 159 and subsequent) assessing only 1 study with limited number of participants. If not bias assessment, at least indicate the statistical power of the studies listed as “best diagnostic performance”
- page 12 lane 240-247. It is not clear to me the significance of some data, referring only to one article (Torres et al., 2012). Some miRNA shows the same behaviour in EM invasion < 50% and > 50%, so they are not of particular clinical significance. In a systematic review, these data are not important.
- page 12 lane 248-250. A “no change” is reported, referring to EM invasion. It is redundant. Probably the whole paragraph about EM invasion can be expunged (see point 2 above) as it is referred only to 1 article
- Figure 2: unreadable, increase letter size
Discussion:
- it could be useful to add a specific paragraph comparing expression of the 33 miRNA found in these studies and a miRNA-cancer database such as dbDEMC? The authors should consider this
Minor points:
- English proof-read: consider a revision from a native-English speaker. The text is difficult to read and some French words are present. a more accurate proof read is needed.
- Figure 1 needs better editing, some boxes hide the last sentences of the inside captions. Moreover, the text mentioned MEDLINE as a research database, not reported in the box; either expunge from the text or add in the box
Round 2
Reviewer 1 Report
This revised manuscript has been improved in the presentation of Results and Discussion sections, so that the concerned context would be more clear. I would recommend that this manuscript to be published with this revised version.
Reviewer 2 Report
The authors have clearly answered to the majority of my comments and I am glad to observe that the quality of the manuscript has been improved by the changes made by the authors.
I have only a minor concern, regarding page 12 lane 240-247, and specifically MI. As the authors have stated in their response, MI has been evaluated in few studies and they cite only one, with contrasting report. I am convinced that this paragraph is confusing and does not add information. I would like to expunge this paragraph.
apart from this, is a good work that systematize the current knowledge of literature and I praise the authors for the manuscript.
